# A Thermophile-Fermented Compost Modulates Intestinal Cations and the Expression of a Juvenile Hormone-Binding Protein Gene in the Female Larvae of Hercules Beetle *Dynastes hercules* (Coleoptera: Scarabaeidae)

**DOI:** 10.3390/insects14120910

**Published:** 2023-11-27

**Authors:** Futo Asano, Taira Miyahara, Hirokuni Miyamoto, Hiroaki Kodama

**Affiliations:** Graduate School of Horticulture, Chiba University, 1-33 Yayoi-cho, Inage-ku, Chiba 263-8522, Japan; afua1049@chiba-u.jp (F.A.); miyahara@chiba-u.jp (T.M.); h-miyamoto@faculty.chiba-u.jp (H.M.)

**Keywords:** compost, fat body, insect, intestinal pH, juvenile hormone, larval growth enhancement, potassium ion

## Abstract

**Simple Summary:**

Various rhino and stag beetles are popular as pets, and their large adults are especially attractive. The female Hercules beetle larvae reared in humus supplemented with a thermophile-fermented compost showed improved growth. Feeding with compost increased the concentration of midgut potassium ions, which was associated with the production of a fermentative unidentified organic acid. Interestingly, adding compost to the humus induced the expression of a gene encoding a hemolymph juvenile hormone-binding protein. Our results suggested that a diet containing fermentative bacteria sex-dependently controls the hormonal condition of saprophagous beetle larvae.

**Abstract:**

The Hercules beetle larvae grow by feeding on humus, and adding a thermophile-fermented compost to the humus can upregulate the growth of female larvae. In this study, the effects of compost on the intestinal environment, including pH, cation concentrations, and organic acid concentrations of intestinal fluids, were investigated, and the RNA profile of the fat body was determined. Although the total intestinal potassium ions were similar between the larvae grown without compost (control larvae) and those with compost (compost larvae), the proportion of potassium ions in the midgut of the compost larvae drastically increased. In the midgut, an unidentified organic acid was the most abundant, and its concentration increased in the compost larvae. Transcriptome analysis showed that a gene encoding hemolymph juvenile-binding protein (JHBP) was expressed in the compost female larvae and not in the control female larvae. Expression of many genes involved in the defensive system was decreased in the compost female larvae. These results suggest that the female-specific enhancement of larval growth by compost was associated with the increased *JHBP* expression under conditions in which the availability of nutrition from the humus was improved by an increase in potassium ions in the midgut.

## 1. Introduction

Various rhino and stag beetles are popular as pets, and their large adults are especially attractive. Because the adult body size of holometabolous beetles is determined based on the body size of juvenile larvae just before the pupal stage [1,2], breeding techniques that improve larval growth have long been investigated. For example, xylophagous larvae of the stag beetle are frequently reared in a sawdust medium infested with mushroom hyphae. This breeding technique greatly benefits the larval growth of stag beetles and is very popular among Japanese breeders of stag beetles. Regarding scarab beetles such as Hercules and Japanese rhino beetles, their larvae are saprophagous, and fermented humus is preferred for larval growth [3]. Recently, it was shown that amending a thermophile-fermented compost to humus improved the female larval growth of the Hercules beetle *Dynastes hercules* [4]. Therefore, beetle larval growth can be enhanced using fermented sawdust and humus, suggesting that larval growth of rhino and stag beetles is accelerated when microbes inhabiting the environment and/or gut microbes degrade the dietary woody biomass. Additionally, it is possible that gut microbiota stimulates the physiological responses in the host larva because of the amendment of a thermophile-fermented compost to humus’ sex-dependently enhanced larval growth [4].

Unlike the mammalian gut, beetle guts are extremely alkaline [5,6,7]. Woody residues swell under alkalinity, reducing cellulose polymerization and crystallinity [8]. The availability of woody residues in the diet correlated with the gut alkalinity of beetle larvae [9]. The gut luminal high alkalinity is simply explained by excess potassium ions [10]. The swollen woody residues are then fermented by gut microbes and converted into organic acids, such as formate, acetate, propionate, and lactate [6,11]. These organic acids are then used as energy for larval growth [12]. In xylophagous termites, acetate is a dominant metabolite of cellulose and hemicellulose, and a major source of termite respiration [13]. Therefore, the intestinal concentration of potassium ions and the production of organic acids from the dietary woody residues are possible key factors determining larval growth.

In insects, two hormones—juvenile hormone (JH) and ecdysteroids—regulate developmental transition (molt and metamorphosis) and larval growth duration. In holometabolous insects, a primary active ecdysteroid, 20-hydroxyecdysone (20E), induces larval–larval molting in the presence of JH [14]. In the final larval instar, 20E induces larval–pupal molting after JH level reduction, indicating that JH prevents larvae from undergoing precocious metamorphosis [14]. JH is produced by the corpora allata, and ablation of the corpora allata reduces the larval growth rate in *Drosophila* [15]. In the tobacco hornworm *Manduca sexta*, the body size is controlled by the timing of JH decay and ecdysteroid secretion [16]. Since the diet quality affects the body size, it is possible that diet modulates the JH level or the timing of JH decay [16]. Additionally, sex-dependent JH effects have been reported in several insects [17,18]. A positive correlation between male adult size and JH level at the prepupal stage was found in the metallic stag beetle *Cyclommatus metallifer* [17]. The adult Japanese mealybug *Planococcus kraunhiae* shows sexual dimorphism, namely, juvenile-like female and winged male. Wing formation in males depends on high JH levels in the larvae at the second instar and prepupal stages, whereas such high JH levels were not observed in female larvae [18]. There are no reports regarding the changes in JH levels in the Hercules beetle larvae.

In this study, compost-induced enhancement of the larval growth rate of the Hercules beetle *Dynastes hercules* was investigated. This compost is made by fermenting marine animal resources at high temperatures by fermentation-associated self-heating [19]. It has been used as a feed additive for laying hens and pigs [20]. In accordance with our previous study [19], the bacterial profile of this compost was considered to consist of mostly Bacillaceae. When Hercules larvae were reared in humus containing 1% (*w*/*w*) compost, an increased growth rate was observed in female larvae [4]. To elucidate the mechanisms of this compost-induced sex-dependent growth rate enhancement, the luminal pH, luminal concentrations of cations, and organic acids were determined in the larvae reared with and without compost. Additionally, the RNA profiles in the fat bodies were determined. The results indicate that humus ingestion with compost increased the potassium ion concentration in the midgut fluid, and a gene encoding the JH-binding protein was expressed in female larvae.

## 2. Materials and Methods

### 2.1. Insects and Rearing Conditions

Domestically bred adults of *D. hercules hercules* (Linnaeus) were purchased from Lumber Jack Co. Ltd. (Koganei, Tokyo, Japan). The humus was purchased from Tsukiyano Mushroom Garden Company (Gunma, Japan). The preparation of the humus bed and rearing conditions of virgin adults have been described in our previous report [4]. After mating, a female adult was reared in a container containing humus. The resulting offspring larvae (eight male and eight female larvae) at the late 2nd or early 3rd instar (100 ± 10 d after hatching) were independently reared in a 2 L-volume blow container filled with humus with or without 1% (*w*/*w*) thermophile-fermented compost and reared for a further 49 d. The control and compost groups consisted of four male and four female larvae. The thermophile-fermented compost contained approximately 40% carbon, 4% nitrogen, 2% phosphorus, 1% potassium, and 4% calcium [21].

### 2.2. Preparation of the Feces, Gut, and Fat Body

When larval weight was measured at 0, 14, 28, 42, and 49 d, the humus on the larvae’s surface was eliminated using a brush, and each larva was weighed. After body weight measurement at 49 d, larvae were placed on sterilized dishes. After defecation, the feces were frozen in liquid nitrogen and stored at −80 °C. After fecal collection, larvae were dissected according to the method of Ogata and Iwabuchi [22]. Each larva was placed in a 100 mL beaker with 5 mL 70% (*v*/*v*) ethanol for 30 min. By exposing the larvae to ethanol vapor, the undigested humus in the foregut was regurgitated. The larvae’s surface was washed using 1% (*v*/*v*) dish detergent (Mama Lemon; Lion Corporation, Tokyo, Japan), and then anesthetized in 100% ethanol for 30 min. Next, the larvae were placed on sterilized filter paper and dissected. The skin on the lateral side of the larval abdomen was cut open along the longitudinal axis using ophthalmic scissors. Fat bodies were removed using tweezers, and the tissues were washed once with phosphate-buffered saline, transferred to a 50 mL centrifuge tube, and stored at −80 °C until use. Next, the entire midgut and hindgut were removed and each organ was stored in a 50 mL centrifuge tube at −80 °C until use.

### 2.3. Determination of pH and Cation and Origanic Acid Concentrations

The frozen midgut and hindgut samples were thawed on ice. The luminal contents were squeezed out and directly subjected to pH measurement. Then, 0.1 g aliquots of gut samples were suspended in 900 μL distilled water and vortexed for 10 min. This mixture was centrifuged at 13,000× *g* for 2 min, and the resulting supernatant was analyzed using ion and liquid chromatographs.

When the cation concentration was determined, the supernatant was filtered using a 0.2 μm syringe filter (GL Sciences Inc., Tokyo, Japan). Cations in the filtrate were quantified using an ion analyzer (IA-300, DKK-TOA Corporation, Tokyo, Japan).

When the organic acid concentrations were determined, the supernatant was filtered with a 0.4 μm syringe filter (Advanced Microdevices Ltd., Ambala, India). The resulting filtrate was passed through Amicon Ultra 0.5 (cut-off molecular weight, 3 kDa; Merck Millipore, Billerica, MA, USA). The organic acid levels in the filtrate were determined using an HPLC Prominence system (Shimazu Corporation, Kyoto, Japan) equipped with a column (Shim-pack SCR-102H).

### 2.4. RNA Extraction and Transcriptomic Analysis

The frozen fat body samples were transferred in a Sepasol-RNA Super-G solution (Nakarai Tesque, Kyoto, Japan), and the total RNA was extracted according to the manufacturer’s protocol. DNase treatment and purification of the resulting total RNA were conducted using the FavorPrep Plant Total RNA Mini Kit (Favorgen Biotech Corp., Ping Tung, Taiwan).

Total RNA from the extracted fat body samples was contracted to Eurofins Genomics (Tokyo, Japan) for mRNA library preparation and sequencing. mRNA was purified as poly(A)^+^ RNA, and paired-end 150 bp sequencing data were generated using a NovaSeq 6000 platform (Illumina, San Diego, CA, USA) (BioProject ID: PRJDB16153, RUN ID: DRR489622-37). Adapter sequences were trimmed, and low-quality reads containing poly-N sequences and/or those below 50 bp were discarded using fastp (version 0.23.2). The read data were trimmed, and de novo assembling using Trinity (version 2.15.1) created the contigs assigned as DN numbers. The resulting contig sequences were assessed using Benchmarking Universal Single-Copy Orthologs (BUSCO version 5.4.7, dataset insect_odb10), and each gene expression level was calculated by aligning the read data for each sample using Bowtie2 (version 2.4.2) and RSEM (version 1.3.3). Cluster analysis was performed using R (version 4.3.1) with normalized expression data as transcripts per million (TPM). The count data were also used to identify differentially expressed genes (DEGs) using edgeR (version 3.42.4). DEGs were annotated using the Basic Local Alignment Search Tool (BLAST version 2.13.0) searches on UniProt (Swiss-Prot and UniProtKB (taxonomy insecta)) data.

## 3. Results

### 3.1. Improved Growth of Female Larvae by Compost

Approximately 100-day-old Hercules beetle larvae were separated according to sex and reared in the humus with or without compost for 49 d (Figure 1). The amendment of the compost to the humus stimulated the growth of female larvae, as previously reported [4], which was observed 14 d after the transfer of larvae to the humus with compost. After 49 d, the relative weights of female larvae reared in the humus with compost increased 2.5-fold compared with the larval weights at 0 d, whereas the weights of larvae reared in the humus without compost increased by 1.8-fold. There was no obvious growth stimulation by compost in male larvae. The larvae were then subjected to the following analyses.

### 3.2. Alkalinity of the Midgut and Hindgut Contents

One possible explanation for the compost-mediated enhancement of female larval growth is the increased availability of woody biomass due to upregulated pH in the intestine of female larvae. The pH of the midgut contents ranged from 8.6 to 9.5, indicating that, as expected, the luminal fluid of the midgut showed alkalinity. The pH of the hindgut fluid was neutral (Figure 2). A slightly increased pH following compost amendment was observed in the midgut of male larvae. However, in the female compost larvae, the pH of the luminal contents of the midgut and hindgut was indifferent compared with those of the control groups. These results indicate that, after the amendment of compost to humus, there were no apparent differences in the pH values of the female intestine compared with the control samples.

### 3.3. Midgut and Hindgut Cation Concentrations

Using ion chromatography, the following cation concentrations in the midgut and hindgut contents were determined: ammonium, calcium, lithium, magnesium, potassium, and sodium ions. Potassium was the most abundant, and its concentration in the midgut of the female compost group increased 2.8-fold compared with that of the control group and reached approximately 8500 ppm. In the hindgut of female larvae, the potassium concentration in the compost group was reduced, whereas that in the control group was markedly increased (Figure 3). Similarly, the potassium ion concentration in the male larvae in the compost group increased in the midgut and decreased in the hindgut compared with those in the control group (Appendix A). These results indicate that the intestinal distribution of potassium ion concentrations between the midgut and hindgut was drastically changed by growth with compost. Interestingly, sodium ion concentrations were specifically increased in the hindgut of female compost larvae. Another remarkable change induced by compost feeding is the intestinal concentration of ammonium ions. In the midgut, approximately 300–400 ppm ammonium ion concentration was detected both in the control male and female larvae, and its concentration was reduced to 5–50 ppm in the compost group (Figure 3 and Appendix A). Therefore, potassium and ammonium ion concentrations were altered in the larvae grown with compost, but no cations showed sex-dependent alterations.

### 3.4. Intestinal Organic Acids

The chromatograms obtained from the HPLC analysis of the intestinal fluid typically showed seven distinct peaks, and six peaks among them were annotated as follows: phosphate, succinate, lactate, formate, acetate, and propionate (Appendix A). The retention time of the largest peak was approximately 17.5 min, similar to that for isobutyrate. However, by increasing the column temperature, it was observed that this peak was separated from that of isobutyrate. Because this peak could not be annotated, we hereafter refer to this unannotated organic acid as RT-17. The RT-17 peak had the largest area in all measured samples and was quantitated using isobutyrate as a standard. The organic acid concentrations in the female and male intestines are shown in Figure 4 and Appendix A, respectively.

The RT-17 concentration, calculated as isobutyrate, was approximately 3000 mg/L in the midgut of the control female group and 4700 mg/L in the compost female group. Subsequently, the RT-17 concentration in the hindgut was drastically reduced by 73% (control group) and 84% (compost group) and reduced to less than 1000 mg/L (Figure 4). The succinate and formate concentrations were markedly reduced in the hindgut of the male and female intestines (Figure 4 and Appendix A). The acetate and lactate concentrations in the female midgut were not different in the control and compost larvae groups. The levels of these two acids in the female hindgut were reduced by 25–50% compared with those in the midgut. In the male intestine, propionate in the hindgut was not detected in the control group, and the acetate concentration level in the hindgut of the compost group was increased compared with that in the midgut of the compost larvae (Appendix A). Possibly, the fermentation of woody biomass was still active in the hindgut of the compost male larvae. In feces, acetate and RT-17 were detected (Appendix A). Fecal levels of RT-17 were similar to those in the hindgut. Fecal levels of acetate were markedly reduced compared with those in the hindgut. There were no sex-dependent differences in organic acid levels in feces.

When the organic acids in the humus were measured, lactate, formate, and acetate were detected at low levels, and none for RT-17 (Appendix A). From these results, RT-17 was the main fermented product in the midgut and was drastically dissimilated or absorbed in the hindgut, implying that RT-17 is vital for the nutrition of Hercules beetle larvae.

### 3.5. Transcriptomic Analysis of Fat Body

To elucidate the mechanisms by which female larvae can efficiently grow by rearing with compost, the mRNA profiles in the fat body were determined. The read data generated from the mRNA sequences of the four groups, i.e., in males and females with compost and in males and females without compost, were 84.6, 84.8, 84.2, and 87.3 million reads, respectively. De novo assembly using these read data resulted in a contig count of 76,145 sequences and a sequence length weighted mean (N50) of 2309 bp. For these assembled data, the BUSCO evaluation using the insect dataset (insect_odb10) yielded 95.0% complete BUSCO, 1.8% fragmented BUSCO, and 3.2% missing BUSCO. The read data were then aligned with the assembled data to obtain the expression level of each sample. Consequently, the alignment rate for all samples was approximately 96%. The dendrogram drawn via hierarchical clustering using the expression data obtained following the alignment with the de novo assembled data suggested that the gene expression profile changed depending on the presence or absence of compost feeding in females (Figure 5). In contrast, no significant difference between the presence and absence of male compost feeding was observed. Therefore, the effects of compost on the mRNA profile were more clearly shown by comparing the mRNA profiles between the compost and control female samples. Each assembled sequence was indicated as the DN number and annotated using the UniProt database.

A total of 110 genes were detected as DEGs, of which 49 were upregulated and 61 were downregulated in the fat bodies of the compost female larvae (Appendix A). The most upregulated gene (DN1122_c1_g1_i4) encodes a hemolymph juvenile hormone-binding protein (JHBP). A downregulated gene (DN1324_c0_g1_i5) in compost female encodes defensin, an inducible antibacterial peptide [23,24]. Two functionally classified categories of downregulated genes were found in compost larvae (Appendix A). One category includes six genes encoding several types of peptidases/proteases. The other category includes seven genes encoding proteins associated with the peritrophic matrix (PM), such as the chitin-binding type-2 domain-containing proteins.

Then, four representative genes, i.e., genes for JHBP, protease (DN2668_c0_g1_i1), PM-associated protein (DN5996_c0_g1_i1), and defensin, were selected, and their expression values that had been calculated as TPM from male and female transcriptome data are shown (Figure 6). JHBP functions as a carrier of JH, and binding of JH to JHBP is necessary to deliver JH to target cells [25,26]. The JH–JHBP complex is delivered through the hemolymph to the target cells. The *JHBP* gene expression was detected in male larvae of the control and compost groups. It was not detected in the control female larvae group, but had almost similar levels in the compost female group as male larvae groups (Figure 6). The defensin gene expression level was high in the control female fat body, but reduced to the levels found in the male fat bodies in the female compost larvae (Figure 6). A gene encoding a putative trypsin-like serine protease (DN2668_c0_g1_i1) was preferentially expressed in the control female larvae, and its expression was low in both male and female compost larvae (Figure 6). Of the six genes in the category for peptidases/proteases (Appendix A), four genes (DN5342_c0_g2_i1, DN27482_c0_g1_i1, DN28385_c0_g1_i1, and DN37931_c0_g1_i1) showed expression patterns similar to that in a putative trypsin-like serine protease (Appendix A). PM is a microfibrillar structure found as an interior layer of the intestine tract. Chitins and proteins are the main constituents of PM, and PM itself is unique in insects [27]. Most PM proteins, such as peritrophin, have chitin-binding domains [28,29]. Chitin deacetylase is believed to maintain the mechanical strength of PM [30]. A gene encoding a chitin-binding type-2 domain-containing protein (DN5996_c0_g1_i1) showed a high expression exclusively in the female control larvae, and not in male control/compost larvae or female compost larvae (Figure 6). The other six genes in the PM protein category (Appendix A) also showed similar expression patterns (Appendix A). These results indicate that, in the fat body of female larvae, the *JHBP* gene expression was high after the amendment of compost to humus. Because defensin, peptidases/proteases, and PM proteins are thought to be associated with the defensive system against pathogenic infection, it is possible that the female larvae grew in healthy conditions by rearing with compost.

## 4. Discussion

One notable finding of this study is the alterations in potassium ion distribution across the intestinal tract of beetle larvae after feeding with humus containing compost. The total amount of potassium ions in the intestine (midgut + hindgut) was estimated, and the proportion of midgut and hindgut potassium ions was calculated (Appendix A). The total amounts of potassium ions were slightly higher in the male larval intestine than those in the female larval intestine. However, they were the same in the control and compost larval intestines. In the compost group, 68% and 56% of total potassium ions were localized in the midgut of female and male larvae, respectively. In contrast, only 18% and 13% of total potassium ions were found in the midgut of female and male larvae of the control group, respectively. Therefore, rearing in compost-containing humus upregulated the midgut’s potassium concentration, especially in female larvae. Because gut luminal alkalinity is established by an excess of potassium ions [10] and there is a correlation between gut alkalinity and larval growth [9], maintenance of high potassium ion concentration in the midgut is possibly involved in the acceleration of humus degradation, which is partly associated with the compost-mediated growth improvement of female larvae. The gut epithelium and Malpighian tubules regulate potassium ion concentration through trans-epithelial transport from the hemolymph to the gut lumen and vise versa [31,32,33]. The effects of compost on the function of potassium ion transport in the gut epithelium and Malpighian tubules remain to be elucidated.

Because the humus used in this study barely contained ammonia (Appendix A), ammonia wass produced in the midgut and reached a concentration of 300–400 ppm in the control larvae, irrespective of sex. In contrast, the ammonia level was drastically reduced in the midgut of larvae grown in the humus with compost (Figure 3 and Appendix A). In the saprophagous larvae of *Pachnoda* spp., the peptides in the humus were degraded in the midgut, and the resulting amino acids are assimilated by the host or gut bacteria. The unassimilated amino acids are then fermented into ammonia, and a high level of ammonia was observed in the hindgut [6]. Because the number of amino acid-fermenting bacteria is low in the midgut of *Pachnoda* spp., the main organ for ammonia production through amino acid fermentation is considered to be the hindgut [6]. In contrast to *Pachnoda*, the high ammonia level in the midgut fluid of the control larvae of the Hercules beetle suggests that ammonia production through amino acid fermentation mainly occurs in the midgut. Conversely, ammonia in the midgut of the compost larvae group showed markedly low levels in female and male larvae, suggesting that the efficiency of amino acid absorption by the host and/or gut bacteria is improved by feeding the humus with compost. Because ammonia uptake by the midgut epithelium has been reported in the tobacco hornworm *Manduca sexta* [34], it is also possible that compost-mediated enhancement of ammonia uptake by the host accounts for the low level of ammonia in the intestine. Interestingly, the ammonia concentration in the feces of domestic pigs was also reduced after supplementation with compost [20]. The catabolized waste of nitrogen compounds in most insects is uric acid. In termites, uric acid is secreted into the hindgut, converted into ammonia, and eventually assimilated by the gut microbiota [35]. Because the ammonia level was decreased in the hindgut of the control larvae of the Hercules beetle, ammonium assimilation likely occurred in the hindgut.

In saprophagous beetle larvae, fermentative degradation of polysaccharides in the humus is an important pathway for host nutrition [36]. Although short-chain fatty acids such as acetate, propionate, and butyrate are the main fermentative products of carbohydrates in the mammalian intestine, acetate is exclusively the dominant fatty acid in the intestine of the saprophagous beetle larvae and xylophagous termites [7,36]. In termites, acetate accounts for the main substrate of respiration [7]. Although acetate is also a dominant fatty acid in the Hercules larval intestine, its concentration was almost the same in the midgut and hindgut fluids (Figure 4 and Appendix A), suggesting that acetate plays a limited role in host nutrition. In addition to acetate, another dominant organic acid, RT-17, may account for the main nutrition source of Hercules beetle larvae. Because RT-17 increased by approximately 1000–1500 ppm in the midgut fluids of male and female compost larvae, it is possible that the potential pH increase due to increased potassium ion concentration observed in the midgut of the compost larvae may be neutralized by the increased RT-17 concentration.

It has been indicated that defensin and serine proteases comprise the innate immune system of beetles. These proteins can be produced by injecting pathogenic bacteria into the insect body. PM protects the intestine of host insects from pathogenic invasion and prevents damage caused by plant-derived anti-nutrients [37]. The genes encoding PM-associated proteins and defensive proteins, such as defensin, and proteases, were preferentially expressed in fat bodies of female control larvae, and their expression was reduced in compost larvae. These results suggest that female larvae grown in the humus with compost remained healthy, probably associated with improved growth. Another possibility is that increased JH levels downregulate these defensive genes. Most genes encoding defensin, proteases, and the PM protein (Figure 6, Appendix A) had low expression levels in male larvae and compost female larvae, whereas the *JHBP* gene expression level was high. The latter possibility remains to be studied. 

Noteworthy, that the expression of the *JHBP* gene was higher in the fat body of compost female larvae than in the control female larvae. The *JHBP* gene is mainly expressed in the larval fat body of the Cotton Bollworm *Helicoverpa armigera* [38] and bamboo borer *Omphisa fuscidentalis* [26]. In lepidopteran larvae, over 99% of JH molecules exist as a JH–JHBP complex [39]. The expression of the *JHBP* gene was consistent with the changes in JH titer, and it was upregulated in the larvae in the rapid growth stage [38]. Therefore, the compost-induced upregulation of *JHBP* expression suggests the consistent upregulation of JH titer in the female Hercules beetle larvae, which would eventually contribute to the accelerated growth rate of female larvae. Finally, our previous report showed the alteration of fecal microbiota by rearing in the humus with compost. In female larvae, the bacteria belonging to the class Mollicutes were specifically increased by compost [4]. Investigating the relationship between compost-induced alteration in gut microbiota and *JHBP* expression or change in JH titer is worthwhile. 

## 5. Conclusions

The amendment of the humus with compost and subsequent feeding of the humus by Hercules beetle larvae modulated the distribution of potassium ions in the larval intestine. Fermentation/degradation of humus is accelerated by upregulated potassium ion levels in the midgut of larvae of the compost group, irrespective of sex, resulting in the production of an unidentified organic acid. This unidentified organic acid is considered to be rapidly assimilated by the host or gut microbiota in the hindgut. The RNA profiles of the fat body showed that the *JHBP* gene was expressed in male larvae of the control and compost groups and in female larvae of the compost group. These results suggest that the female-specific enhancement of larval growth was associated with the increased expression of the *JHBP* gene under conditions in which the availability of nutrition from the humus was improved by an increase in potassium ions in the midgut.

## Figures and Tables

**Figure 1 insects-14-00910-f001:**
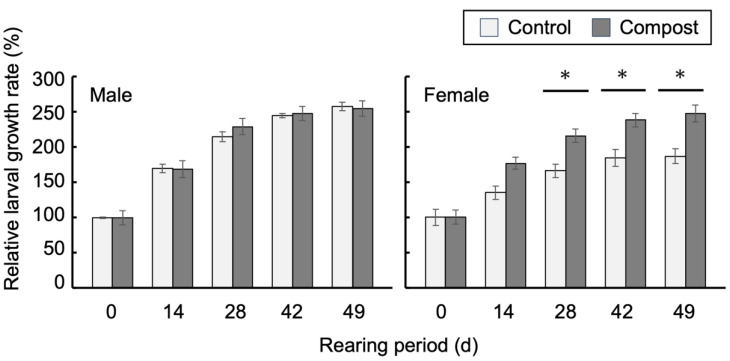
The amendment of a compost to the humus stimulates the growth of female larvae of the Hercules beetle. Larvae were transferred into the humus with or without compost (0 d) and then reared for 49 d. The relative values of the mean body weight at the start of the experiment are shown. Bars represent SD. * *p* < 0.05. *n* = 4.

**Figure 2 insects-14-00910-f002:**
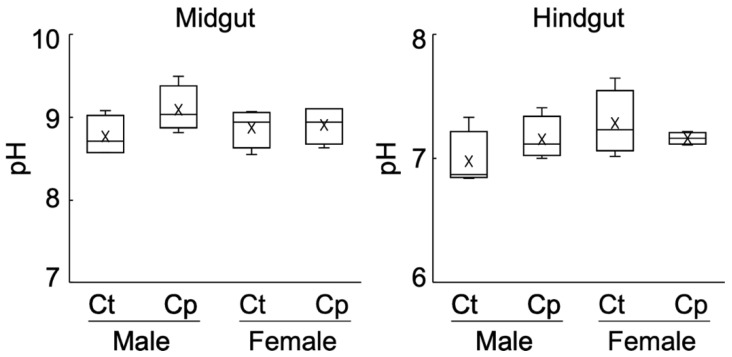
pH of intestinal fluids of the Hercules beetle larvae. Ct and Cp indicate the control and compost groups, respectively. The middle line is the median, and “x” is the mean. The top and bottom ends of the whiskers show the maxima and minima of the data, respectively. *n* = 4.

**Figure 3 insects-14-00910-f003:**
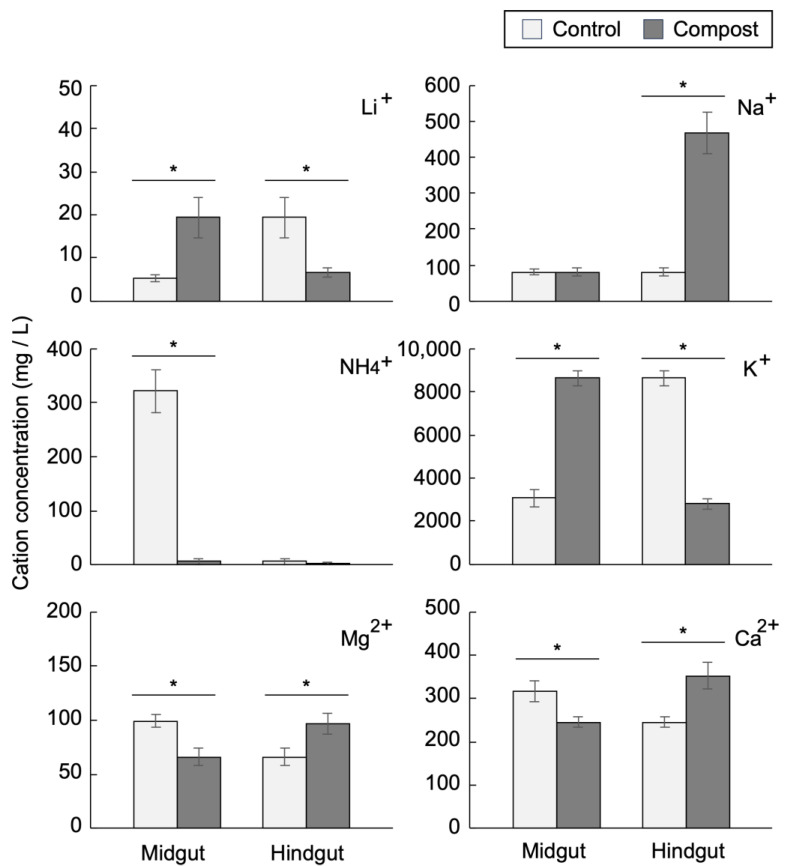
Cation concentrations in intestinal fluids of female Hercules beetle larvae. Bars represent SD. * *p* < 0.05. *n* = 4.

**Figure 4 insects-14-00910-f004:**
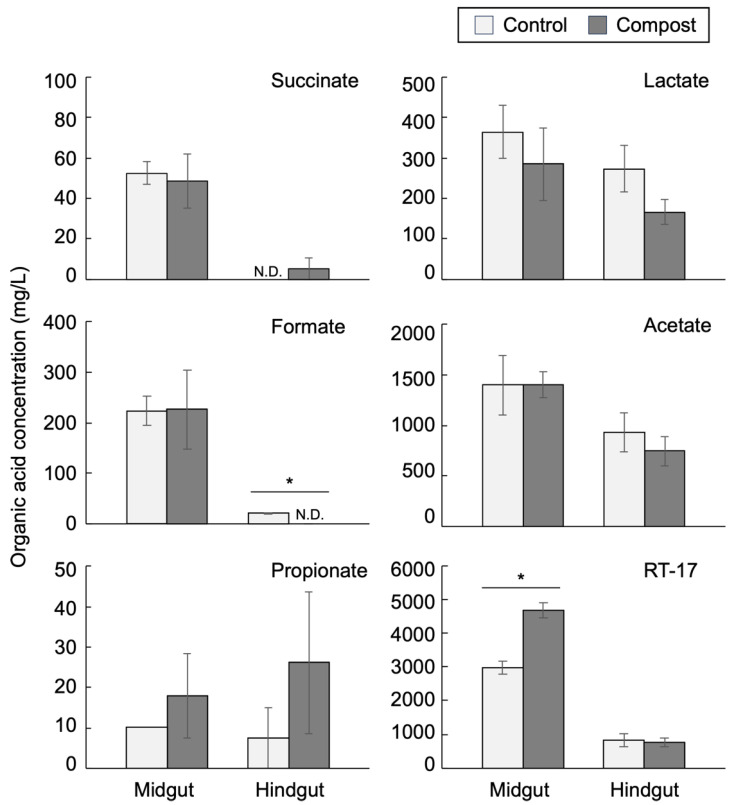
Organic acids in the female intestinal fluids. RT-17 is an unidentified compound with a retention time of approximately 17 min. N.D. represents not detected. Bars represent SD. * *p* < 0.05. *n* = 4.

**Figure 5 insects-14-00910-f005:**
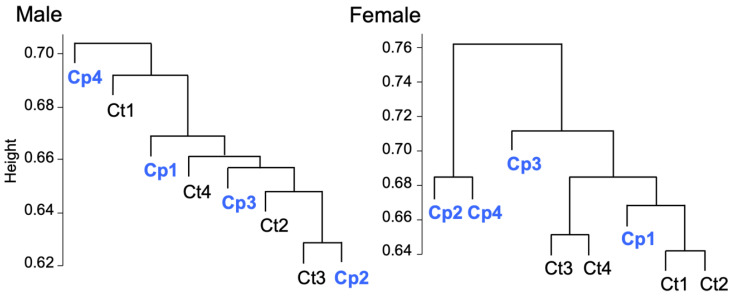
Hierarchical cluster tree of genes expressed by male and female Hercules beetle larvae. Four larvae reared in the humus without compost (Ct1–4) and four larvae reared in the humus with compost (Cp1–4) were subjected to transcriptome analysis. Dendrogram generated from 74,571 genes expressed in at least one of the eight samples for each sex.

**Figure 6 insects-14-00910-f006:**
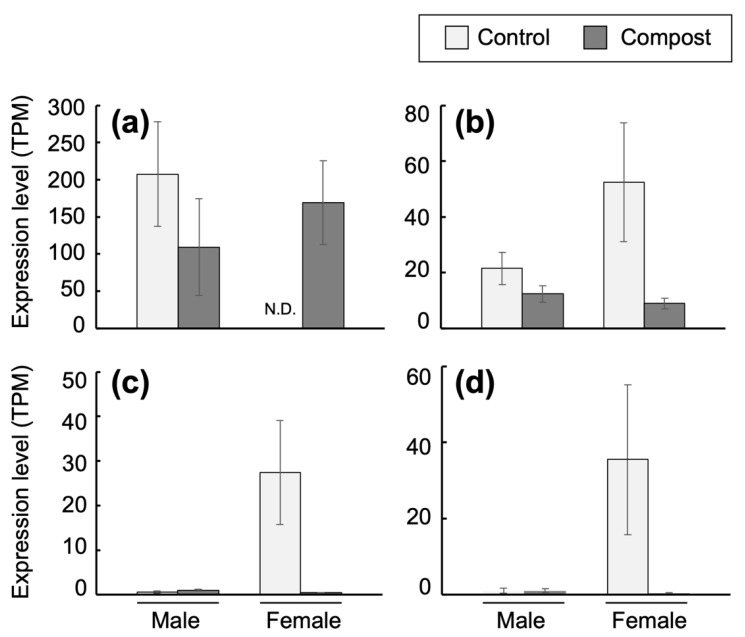
Expression patterns of four representative DEGs that were found in the comparison between the female control larvae and compost larvae. DEGs are listed in Appendix A. The data collectively show the transcripts per million mapped reads (TPM) obtained from the male and female larvae. Data show the expression patterns of the following genes: (**a**) hemolymph *JHBP* gene (DN1122_c1_g1_i4), (**b**) defensin (DN1324_c0_g1_i5), (**c**) trypsin-like serine protease (DN2668_c0_g1_i1), and (**d**) chitin-binding type-2 domain-containing protein (DN5996_c0_g1_i1). Bars represent SD. *n* = 4.

## Data Availability

The transcriptomic data have been deposited into BioProject ID: PRJDB16153, RUN ID: DRR489622-37.

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
