# Peer review of "A Thermophile-Fermented Compost Modulates Intestinal Cations and the Expression of a Juvenile Hormone-Binding Protein Gene in the Female Larvae of Hercules Beetle Dynastes hercules (Coleoptera: Scarabaeidae)"

_insects, 2023, doi:10.3390/insects14120910_

Round 1
Reviewer 1 Report
Comments and Suggestions for Authors
REVIEWER’S COMMENTS:
This manuscript presented research of the effects of feeding on humus, and adding a thermophile-fermented compost to the humus on intestinal environment, including analysis of pH, cation, and organic acid concentration of intestinal fluids and the expression of a juvenile hormone-binding protein gene in the female and male larvae of Hercules beetle Dynastes hercules (Coleoptera: scarabaeidae).
The research is interesting and presents promising results. This is a review on an important topic that has developed rapidly in the last years. It fits perfectly in the scope of Pathogenes. The authors have included and analysed many references and my (quick) literature search identified no gaps.
Line 71: “Additionally, sex-dependent JH effects have been reported in several insects. Please add literature.
If JH sex-dependent effects have been also reported in Hercules beetles? If not, please write it.
Line 81-82: “The bacterial profile of this compost showed that most belong to Bacillaceae”
You did not analyze the bacterial profile of this compost in this study. So, you can write that you previous study or reasearch others Researchers indicated that identified bacteria the most belongs to Bacillacea, but you can not write that this this compost showed that most belong to Bacillaceae.
You can also write “In accordance with available literature data…..
In the section 2. Materials and Methods should be well described study groups. How many research groups were in the study? what constituted control groups? How many repetitions were in the groups?
It is not clear.
Line: 93-94: “The preparation of the humus bed and rearing conditions of virgin imagoes have been described in our previous report [4].” It should be described in the paper.
In my pion of view, the tables should be presented in colors and whole gene expression in plots. It will more interesting.
Line 266-267 The authors cite literature that does not contain such data: A downregulated gene (DN1324_c0_g1_i5) in compost female encodes defensin, an inducible antibacterial peptide.
Ishibashi, J.; Saido-Sakanaka, H.; Yang, J.; Sagisaka, A.; Yamakawa, M. Purification, cDNA cloning and modification of a defensin from the 468 coconut rhinoceros beetle, Oryctes rhinoceros. Eur. J. Biochem. 1999, 266, 616–623. (doi: 10.1046/j.1432-1327.1999.00906.x)
There is only the sentence: ”Midguts of O. rhinoceros larvae may be constantly infected by bacteria, because they feed on dung of cattles and grow in compost”. What means that bacteria induce expression of defensins.
The tables are not included with the manuscript.
Reviewer 2 Report
Comments and Suggestions for Authors
The paper is an attempt to dissect molecular mechanisms underlying impact of substrate fermentation on the physiology of insect larvae inhabiting (and feeding upon) humus. The insect studied is an interesting research model and compost fermentation is a widely occurring phenomenon, so the paper is of interest for the readers of the journal and perfectly suits its scope. The experimental design is concise and straightforward, while the methods are complex, including biochemical and molecular genetic tests. The work follows a series of preceding studies and the results are logically explained. Attention of the authors to insect sex and differential assays allowed discovering some gender-dependent processes. Discussion gives insights into alternative possible explanations and sound convincing. The paper is clearly written, typos and other mistakes are few. A list of minor flaws is given below and minor revision is recommended.
L9 and elsewhere: “imago” is not a common term in scientific English literature, “adult” is preferred
L19-20: you use “organic acid concentration” as singular here but plural in the main text
L24-26: does it mean that in control larvae, this gene is not expressed?
L51: I do not see a reason to refer to mammalian gut
L67-68: I doubt that this phrase is necessary here, what’s the need for this?
L69: I’m not sure I fully understand the meaning and the syntaxis of “plasticity … by diet quality is determined by … rate”
L82: what “most” stands for?
L122: is plural needed for “paper”?
L140: data was = data were
L170: consider using synonyms to avoid repetition of the same word within a single line of text (“increased”)
L256: I didn’t find an explanation of what “DN” stands for
L313: caluculated = calculated
L325: eptithelial = epithelial
L262 and elsewhere: sexuality may not be a proper synonym for “sex” or “gender”
Comments on the Quality of English Languagesmall typos and mistakes are listed above, overall quality is good, the paper is well written
